# Comparison of Anatomic and Non-Anatomic Liver Resection for Hepatocellular Carcinoma: A Retrospective Cohort Study

**DOI:** 10.3390/medicina58091305

**Published:** 2022-09-19

**Authors:** Elvan Onur Kirimker, Alp Togan Kirac, Suleyman Utku Celik, Can Yahya Boztug, Muharrem Berat Kaya, Deniz Balci, Mehmet Kaan Karayalcin

**Affiliations:** 1Department of General Surgery, Ankara University School of Medicine, 06230 Ankara, Turkey; 2Department of General Surgery, Ankara Training and Research Hospital, 06010 Ankara, Turkey; 3Department of General Surgery, Gulhane Training and Research Hospital, 06000 Ankara, Turkey; 4Department of General Surgery, Division of Surgical Oncology, Abdurrahman Yurtaslan Ankara Oncology Training and Research Hospital, 06200 Ankara, Turkey; 5Department of General Surgery, Bahcesehir University School of Medicine, 34353 Istanbul, Turkey

**Keywords:** hepatocellular carcinoma, liver resection, outcomes, surgery

## Abstract

*Background and Objectives*: The survival benefit of anatomical liver resection for hepatocellular carcinoma has not been elucidated yet. In this study, we aimed to investigate the effects of anatomic and non-anatomic liver resection on surgical outcomes in patients with hepatocellular carcinoma. *Materials and Methods*: A retrospective analysis of patients undergoing anatomic or non-anatomic resections due to hepatocellular carcinoma between March 2006 and October 2019 was conducted. Demographics, preoperative laboratory assessments, treatment strategies, and postoperative outcomes were analyzed. *Results*: The total cohort consisted of 94 patients, with a mean age of 63.1 ± 8.9 years, and 74.5% were male. A total of 41 patients underwent anatomic liver resection, and 53 patients underwent non-anatomic resection. The overall survival rates were found to be similar (5-year overall survival was 49.3% for anatomic resection and 44.5% for non-anatomic resection). Estimated median overall survival times were 58.5 months and 57.3 months, respectively (*p* = 0.777). Recurrence-free 1-, 3-, and 5-year survival rates were found to be 73.6%, 39.1%, and 32.8% in the non-anatomic resection group and 48.8%, 22.7%, and 22.7% in the anatomic resection group, respectively. Grade three or higher complication rates were found to be similar among the groups. *Conclusions*: This study did not find a difference between two surgical methods, in terms of survival. A tailored selection of the resection method should be made, with the aim of complete removal of tumoral lesions and leaving a suitable functional liver reserve, according to the parenchymal quality and volume of the liver remnant.

## 1. Introduction

Surgical resection is preferred curative treatment for patients with a limited number of small tumors and preserved liver function in the treatment of hepatocellular carcinoma (HCC). In the setting of decompensated cirrhosis, liver transplantation is the only curative option for suitable patients, according to the guidelines [1]. Surgical procedures with curative intent should be considered after proving absence of extrahepatic spread. While emerging medical therapies are shown to improve the survival of patients with metastatic or unresectable liver cancer, liver resection is the gold standard for suitable patients [2,3]. Anatomic liver resection was described as the systematic removal of a hepatic segment confined by tumor-bearing portal tributaries [4]. This approach has been advocated to have superior outcomes to non-anatomic liver resections, in terms of recurrence and survival [5,6]. This procedure is usually achieved by compressing, clamping, or injecting stains into relating portal branches to make the margins of the portal territory visible. Anatomic resection is expected to remove potential microportal invasion and micrometastases in the tumor-bearing segment of the liver. It is also claimed to be beneficial for having wider tumor-free margins. The superiority of the anatomic resection over non-anatomical resection for HCC remains a matter of a debate. Despite the large multicentric studies and meta-analyses, this topic needs to be addressed [6,7,8,9,10]. In this study, we hypothesized that anatomic resection would result in decreased recurrence and better survival outcomes, compared to the non-anatomic resection.

## 2. Materials and Methods

### 2.1. Study Design, Study Population, and Data Collection

This was a retrospective cohort study comparing outcomes two surgical resection techniques (anatomic vs non-anatomic) for HCC at Ankara University School of Medicine, a tertiary healthcare hospital, between March 2006 and October 2019. The study was performed in line with the principles of the Declaration of Helsinki. Ethical approvals were obtained from both the Ethics Committee of Ankara University School of Medicine (İ-07-404-22, approved on 22 July 2022). Patients younger than 18 years, those directly underwent liver transplantation, and those were treated with only intraoperative ablative procedures, such as microwave or radiofrequency ablation, were excluded from the study. Stage C patients, according to Barcelona Clinic Liver Cancer Classification (BCLC), are not candidates of surgical resection, and Stage D patients are exceptional candidates for transplantation, as a policy of our institution. Therefore, those two subgroups of patients with HCC were naturally excluded from the study. All the lesions were confirmed as HCC histopathologically. Data collected included age, sex, comorbities, viral and tumor markers, postoperative laboratory values, complications, and readmissions. Clinical notes regarding clinical course of patients, radiologic data relating to tumor, and postoperative pathologic diagnosis were obtained from the database. Long-term follow-up and survival data were collected by review of patients’ clinical records and phone contact with patients or their family members.

Model for end-stage liver disease (MELD) scores of patients were calculated from the last preoperative laboratory results [11]. Stage of the disease of patients were categorized into groups, according to BCLC [12]. Complications were classified according to Dindo-Clavien classification system and ≥ Grade 3 complications were assumed as major [13].

### 2.2. Surgical Technique

Anatomic segmentectomies, sectionectomies, hemihepatectomies, and trisectionectomies were recorded as anatomic procedures, unless those were performed by following tumor margins, rather than following anatomic landmarks or portal demarcation lines. Illustration shows anatomic and non-anatomic resection lines in different tumor locations (Figure 1). Anatomic operations were classified according to Brisbane nomenclature [14]. All operations were performed by experienced surgeons dedicated on hepatobiliary and transplant surgery. Pedicle clamping for performing anatomic resection is the method of choice in the department (Figure 2). Lesions in cirrhotic livers of patients with portal hypertension were the main reason for choosing non-anatomic resections, with the intent of preserving more liver parenchyma. Preference of surgical approach, which is usually based on abovementioned principles, were decisions of patient’s surgeons.

### 2.3. Statistical Analysis

Collected data were anonymized prior to analysis. Study cohort was divided into two groups: anatomic and non-anatomic resection. Continuous variables were presented as mean ± standard deviation or median and range. For dichotomous data, we presented frequencies and percentages. Primarily two groups were compared for differences between recurrence and survival. Student’s *t*-tests and Wilcoxon Mann–Whitney U-tests were used to compare continuous variables. χ2 and Fisher exact tests were used to compare categorical variables. Kaplan–Meier survival curves were created, and the log-rank test was employed to compare overall survival between patients undergoing anatomic and non-anatomic resections. All analyses were performed with Statistical Product and Service Solutions (SPSS) for Macintosh, version 16.0 (SPSS Inc., Chicago, IL, USA). All tests were two-sided, and a *p*-value < 0.05 was considered statistically significant.

## 3. Results

A total of 94 patients with a mean age of 63.1 years (range, 25–82) were included in the study. The male:female ratio was 2.9, indicative of a male dominance. While 41 (43.6%) patients underwent anatomic liver resection, 53 (56.4%) patients underwent non-anatomic resection. Age, preoperative laboratory values, MELD scores, alpha-fetoprotein levels, comorbidities, and hepatitis B virus (HBV) or hepatitis C virus (HCV) positivity were similar between the anatomical and non-anatomical resection groups. Cirrhosis and BCLC stage 0-A disease was less frequent in the anatomic resection group than the non-anatomic resection group (*p* = 0.008), while maximum tumor diameter was significantly larger in the anatomic resection group (*t*_91_ = 2.032, *p* = 0.045) (Table 1). Postoperative 5th day total bilirubin level was found to be lower in the non-anatomic resection group, indicating the recovery of liver function (*t*_85_ = 2.062, *p* = 0.022). Postoperative 5th day international normalized ratio (INR) was similar between the groups. The length of hospital stay of the patients undergoing non-anatomic resection was significantly shorter (*t*_86_ = 2.861, *p* = 0.006). The major complication rates were similar. Microvascular invasion was more frequently reported in the anatomic resection group (*p* = 0.028) (Table 2).

Kaplan–Meier curves were created for comparison of overall and recurrence-free survival rates (Figure 3). Overall survival rates were found to be similar. Five-year overall survival was 49.3% for the anatomic resection group and 44.5% for the non-anatomic resection group. The estimated median overall survival times were 58.5 months and 57.3 months (*p* = 0.777), respectively. Recurrence-free 1-, 3-, and 5-year survival rates were found to be higher in the non-anatomic resection group than the anatomic resection group. However, these differences did not reach statistical significance (*p* = 0.085). Moreover, the extrahepatic recurrence rate was found to be higher in the anatomic resection group (*p* = 0.020). The estimated median recurrence-free survival times were 11.5 months in the atomic resection group and 23.7 months in the non-anatomic resection group (*p* = 0.085). The 1-, 3-, and 5-year disease-free and overall survival rates of patients are presented in Table 3 and Table 4.

Number of anatomic and non-anatomic resections for HCC at the institute and trend of increasing practice of anatomic resection are shown (Figure 4).

## 4. Discussion

Anatomic resection has been thought to be superior in the surgical treatment of HCC. However, the supporting evidence is insufficient. This study did not reveal a significant difference between anatomical and non-anatomical resection for HCC, in terms of survival. Complication rates were also found to be similar. These results may be a reflection of the proper selection of treatment modality, rather than the utilization of same surgical approach universally for every type of local presentation of the disease because the survival benefits of anatomic and non-anatomic resections for HCC is still debated [15]. The main basis of the hypothesis of the advantage of anatomical resection is the assumption that the main blood flow of the tumor is supplied by hepatic artery, and reverse flow of the portal vein is responsible of tumor spread into the neighbor portal branch [16]. Thus, the removal of the whole portal unit containing the probable micrometastases may be advantageous, mainly in terms of preventing local recurrence, which is usually seen two years postoperative and through improving survival. On the other hand, only the mass lesion that is covered with liver parenchyma, with an acceptable thickness of usually 1 cm, is removed in non-anatomic liver resections. Non-anatomic resections, regardless of the thickness of the margins, while targeting the tumor capsula to remain intact, were also reported with comparable outcomes [17]. Considering the two surgical techniques, there is more liver parenchymal loss in the anatomic resections, as the ‘suspect’ or ‘threatened’ tumor surrounding liver tissue is also removed. Chronic liver parenchymal disease or cirrhosis frequently take place in the etiology of HCC. Advocates of non-anatomic resection claim that excess parenchyma removed by anatomic resections may increase the risk of liver failure after surgery.

One of the most frequently used treatment and classification algorithms is BCLC [18]. BCLC, which has been revised with changing and developing treatments after its first version, is also criticized in some respects. Resection treatments are only suitable for Grade 0 and 1 patients, according to BCLC. However, there are publications that report survival results of liver resections similar to those of Grade 0 and A patients in Grade B patients [19,20]. In addition, there are publications reporting long term survival, which can be achieved with liver transplantation in a subgroup of Grade D patients who are recommended only palliative treatment and have a short survival [21]. The vast majority of publications comparing anatomical resections and non-anatomic resections are retrospective publications and the meta-analyses produced from them. Among these publications, there are studies favoring anatomical resection, as well as studies concluding non-anatomic resections are not inferior to anatomical resections, in terms of survival [22,23,24,25,26,27,28,29,30]. Some surgeons may tend to routinely apply one of these two methods. However, both approaches are performed in most centers. In some patients, when one of the segments, or the segments associated with the tumor, is fed through multiple thin portal vein branches, it may not be possible to identify this segment by dye injection or pedicle clamping into the portal vein. In a series reported from Korea, in 7.1% of patients targeted for anatomical segmentectomy, dye injection could not be performed through the portal vein, and anatomical segmentectomy was not possible because the segment associated with the mass was fed by scattered tiny portal vein branches, and the mass could not be seen clearly [31].

After the 1990s, there are a large number of case series comparing anatomical and non-anatomic resections for HCC. Anatomic resection has been reported to bring a survival advantage over non-anatomic resections in many case control series, mostly originating from the far east [27,32,33,34]. The largest of these is the Japanese Nationwide Study, in which 5781 patients are included. In this retrospective study, anatomic segmentectomy has been reported to be advantageous, in terms of disease-free survival, especially in tumors with a diameter of 2 to 5 cm [35]. In another study, it was concluded that anatomic resection might be a better strategy to achieve low early recurrence rates [36]. In some of the publications favoring anatomical resection, patient groups were matched with propensity score matching, and statistical acuity was increased [5,37]. Ishii et al. reported a survival advantage, in favor of anatomical resection, after a propensity score matching in Child–Pugh A patients, with tumors smaller than 5 cm, and by comparing two groups with a mean tumor number of 1.5 [30]. In a case series from Japan that included patients with a single HCC smaller than 5 cm, anatomical resections resulted in better survival in patients with preserved liver function, while non-anatomical resections resulted in better survival in patients with cirrhosis [38]. Moreover, Hwang et al. reported that anatomical resection (AR) had no prognostic advantage over non-anatomical resection (NAR) in patients with solitary HCC > 5 cm with microvascular invasion, according to their experience with 2558 solitary HCC resections [39].

Besides articles that mention the advantages of anatomical resection, there are also propensity score matching studies reporting that there is no difference, in terms of the survival between anatomical and non-anatomical resection [24,28,29]. Meta-analyses of articles comparing two surgical approaches are undoubtedly more valuable, in terms of evidence. In two meta-analyses, anatomical resections have been reported to have better outcomes, in terms of overall survival and disease-free survival [9,40]. Additionally in another meta-analysis, Zhou et al. reported that anatomic resections were found to be superior, in terms of overall and disease-free survival [10]. In two other meta-analyses, better survival in anatomical resections and overall survival, similar to non-anatomic resections, were reported [7,41]. Tan et al. also favors anatomic resections, in terms of 5-year survival and recurrence in their meta-analysis [8]. Jiao et al. reported better recurrence free and overall survival rates for patients underwent AR than patients underwent NAR in their meta-analysis, including 9120 patients from 38 articles [23].

In this study, providing the data from a single center made the sample more homogeneous, in terms of surgical technique, indications, and anesthesia. Although long-term follow-up results were obtained, the sample size is a limitation of the study. It can be considered that the outcomes of a patient group with a long-term follow-up, in this way, is a valuable data source for meta-analyses in which the studies on the subject are discussed. The current study evaluates the contribution of anatomical resection to survival and provides information on the etiology of HCC and results of surgical resections. HBV positivity, which has an important place in the etiology of cirrhosis in Turkey, is also frequently seen in HCC patients. Due to their early diagnosis at follow up visits in our institute for chronic liver disease, some patients were operated on with a small single tumor, while patients who were sent directly from other hospitals with the diagnosis of large tumors enabled the evaluation of patients with a wide spectrum of tumors in size and number. The high HBV positivity in the study population is compatible with the place of HBV in the etiology of cirrhosis in our country.

One of the important parameters that was investigated by the present study was the recurrence rate. In our cohort, recurrence rates were found to be 58.5 % (24 patients) in the anatomic resection group, while it was 34% (18 patients) in the non-anatomic resection group. Extrahepatic presentation of recurrence was also more frequent in the anatomic resection group. When we evaluated this situation, we thought that one of the primary reasons was the tumor size and microvascular invasion rate in the anatomic resection group, which were significantly higher than that of the non-anatomic resection group. This can be translated into occult metastases, which can be expected to be more common in the anatomic resection group, that include larger primary lesions with significantly larger diameters that lead to extrahepatic recurrences. The rate of macrovascular invasion was also higher in the anatomic resection group, although no statistically significant difference was found. In addition, the rate of margin positivity in the anatomic resection group was lower than the non-anatomic resection group, as expected, which strengthened the argument that the recurrence rate was higher due to more aggressive tumor biology. In a study by Shindoh et al., the tumor size and microvascular invasion rate were found to be higher in the anatomic resection group, compared to the non-anatomic resection group, while the local recurrence rate was found to be lower in the anatomic resection group; in this study, tumor size >2 cm, İndocyanine green clearance rate at 15th minute (ICG-R15) >15%, and the presence of microvascular invasion were identified as independent predictive factors for other types of recurrence (i.e., nonlocal recurrence) [42].

Estimated median recurrence-free survival time, which is a parameter that correlates with the follow-up times of the patients, was found to be lower in the anatomic resection group. Because the anatomic resection group included patients who were operated on later in the study inclusion timeframe. This caused the patients in the anatomic resection group to have shorter follow-up times than the patients in the non-anatomic resection group; therefore, the parameter that correlated with this situation was also compatible with this result. Another important point is the follow-up after liver resection. Strict follow-up after resection results in timely initiation of proper treatment protocols for recurrence. Therefore, the overall survival rates found were similar, even though the disease-free survival rate of patients in the anatomic resection group were insignificantly lower than that of the patients in the anatomic resection group

The present study results showed that there was no superiority of anatomic resection, compared to non-anatomic resection, in terms of either the recurrence-free or overall survival in patients with HCC. A survival advantage may not be obtained in anatomical resection of a single peripheral lesion that can be removed with a non-anatomical resection with wide surgical margins. In a non-anatomical wide resection of this lesion, the adjacent parenchyma, which is planned to be removed by anatomical resection, and perhaps even more parenchyma, may be removed, even if it is not technically and approachably called anatomical resection. In large single tumors, for example, an HCC that tends to remain single until it reaches a large diameter of 10 cm may also be considered to have a lower tendency to metastasize intrahepatically. Therefore, it can be argued that non-anatomical resection of this lesion without entering the tumor can provide the same survival benefit as anatomical resection in this scenario.

## 5. Conclusions

Anatomical and non-anatomical resections of HCC do not differ in terms of survival, according to the current study conflicting with most of the literature on this topic. The heterogeneity of the tumor size and quality of liver parenchyma may be a source of bias. Thus, a tailored selection of the method for resection should be made, with the aim of the complete removal of the tumor threat and leaving a suitable functional liver reserve, according to the parenchymal quality and volume of the liver remnant. We recommend anatomic resection for surgical treatment of HCC when it is possible, considering liver function, portal hypertension, and the existence of cirrhosis, based on current literature and our findings. Further meta-analyses are necessary with multicenter, randomized, and prospective studies.

## Figures and Tables

**Figure 1 medicina-58-01305-f001:**
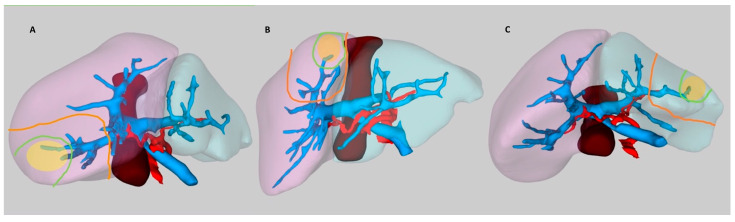
Anatomic (orange line) and non-anatomic (green line) resection lines in different tumor scenarios of the same liver on 3D images. (**A**) Tumor occupying segment 6-7 junction, right inferolateral view; (**B**) Tumor in the Segment 8, anterior view; (**C**) Tumor in the segment 3, inferior view. Right hemiliver (purple), left hemiliver (turquoise), portal vein (blue), hepatic artery (red), vena cava inferior (brown), tumor (yellow). Images were created with Livervision (LiverVision™, Medivision Ltd. Ankara, Turkey).

**Figure 2 medicina-58-01305-f002:**
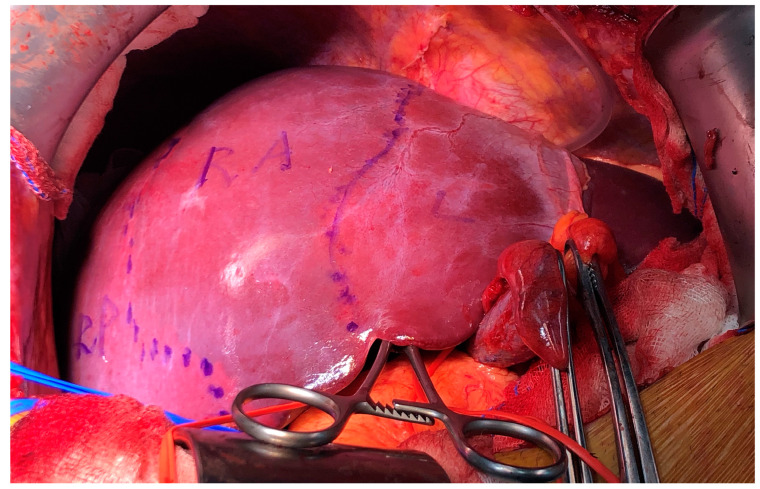
Intraoperative photo of right anterior section demarcation after clamping right anterior pedicle for anatomic resection of a tumor occupying both segment 5 and segment 8 of the liver. RA, right anterior section; RP, right posterior section; L, left hemiliver.

**Figure 3 medicina-58-01305-f003:**
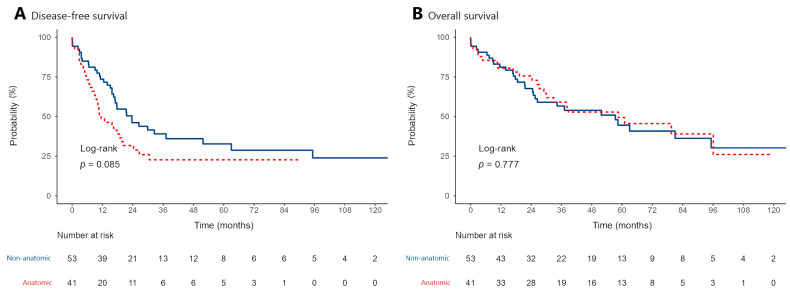
Kaplan–Meier curves showing disease-free (**A**) and overall survival (**B**) patterns of patients who underwent anatomic and non-anatomic liver resections for the treatment of hepatocellular carcinoma (HCC).

**Figure 4 medicina-58-01305-f004:**
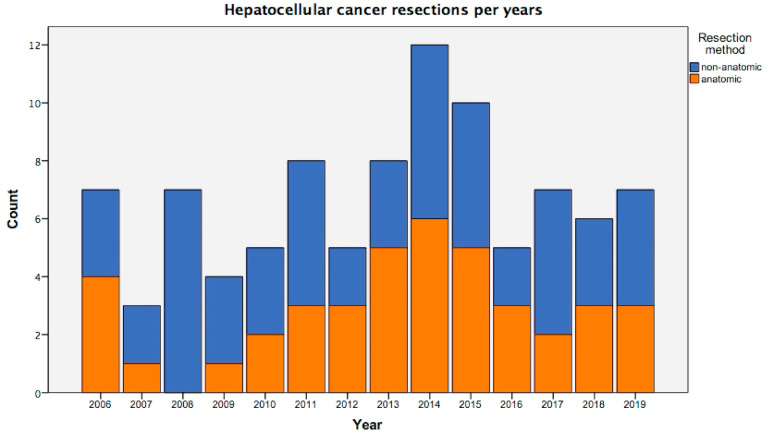
Number of anatomic (orange) and non-anatomic (blue) liver resections those were performed for the treatment of hepatocellular carcinoma at the institute by years.

**Table 1 medicina-58-01305-t001:** Patient and disease characteristics, with comparisons between patients who underwent anatomic and non-anatomic resections.

	Anatomic Resection (*n* = 41)	Non-Anatomic Resection (*n* = 53)	*t*(df) *, *p*-Value
Age (year)	64.6 ± 8.6	61.9 ± 9.0	1.450(92), 0.150
Male gender, *n* (%)	33 (80.5)	37 (69.8)	0.239
Hypertension, *n* (%)	17 (41.5)	16 (30.2)	0.256
Diabetes mellitus, *n* (%)	7 (17.1)	15 (28.3)	0.202
Cardiovascular disease, *n* (%)	8 (19.5)	7 (13.2)	0.408
Pulmonary disease, *n* (%)	0 (0)	4 (7.5)	0.072
Chronic renal insufficiency, *n* (%)	3 (7.3)	3 (5.7)	0.745
HBV positivity, *n* (%)	20 (48.8)	30 (56.6)	0.451
HCV positivity, *n* (%)	7 (17.1)	9 (17.0)	0.991
Cirrhosis, *n* (%)	19 (46.3)	39 (73.6)	**0.007**
Hemoglobin (g/dL)	13.3 ± 1.8	13.6 ± 1.9	0.669(92), 0.505
Platelet counts (×10^3^/mm^3^)	170.7 ± 56.6	157.6 ± 77.2	0.946(91), 0.347
Total bilirubin (mg/dL)	0.87 ± 0.37	0.92 ± 0.50	0.577(87), 0.565
Albumin (g/dL)	3.69 ± 0.68	3.61 ± 0.71	0.503(92), 0.616
INR	1.1 ± 0.1	1.1 ± 0.2	1.885(84), 0.063
Creatinine (mg/dL)	1.05 ± 0.80	0.90 ± 0.25	1.127(43), 0.266
MELD	8.6 ± 3.5	9.2 ± 3.6	0.797(83), 0.428
Alpha-fetoprotein (ng/mL)	51.8 (1.3–54,000)	11.3 (1.5–3402)	**1.765(35), 0.006**
Maximum tumor diameter (mm)	57.4 ± 38.2	42.2 ± 33.9	**2.032(91), 0.045**
Tumor number	1.6 ± 1.4	1.3 ± 0.8	1.091(59), 0.280
BCLC early disease (0-A), *n* (%)	12 (29.3)	30 (56.6)	**0.008**

HBV, hepatitis B virus; HCV, hepatitis C virus; INR, international normalized ratio; BCLC, Barcelona Clinic Liver Cancer; MELD, model for end-stage liver disease; **t* value and degree of freedom (df) is reported for only *t*-test results.

**Table 2 medicina-58-01305-t002:** Postoperative outcomes, with comparisons between patients who underwent anatomic and non-anatomic resections.

	Anatomic Resection (*n* = 41)	Non-Anatomic Resection (*n* = 53)	*t*(df) *, *p*-Value
Postoperative day 5 total bilirubin (mg/dL)	1.77 ± 1.51	1.35 ± 0.98	**2.062(85), 0.022**
Postoperative day 5 INR	1.3 ± 0.2	1.3 ± 0.2	0.332(62), 0.881
Hospital stay, day	12.6 ± 13.4	8.1 ± 7.0	**2.861(86), 0.006**
≥ Grade 3a complication, *n* (%)	6 (14.6)	6 (11.3)	0.633
In hospital mortality, *n* (%)	5 (12.2)	5 (9.4)	0.667
Margin positivity, *n* (%)	7 (17.1)	12 (22.6)	0.505
Microvascular invasion, *n* (%)	18 (43.9)	12 (22.6)	**0.028**
Macroinvasion, *n* (%)	15 (36.6)	10 (18.9)	0.054
Recurrence, *n* (%)	24 (58.5)	18 (34.0)	**0.017**
Isolated hepatic recurrence, *n* (%)	16 (39.0)	16 (30.2)	0.370
Extrahepatic recurrence, *n* (%)	6 (14.6)	1 (1.9)	**0.020**
Synchronous hepatic and extrahepatic recurrence	2 (4.9)	1 (1.9)	0.413
Lung, *n* (%)	3 (7.3)	1 (1.9)	0.196
Bone, *n* (%)	3 (7.3)	1 (1.9)	0.196
Peritoneum, *n* (%)	2 (4.9)	0 (0.0)	0.104
Recurrence treatment			
Best supportive care, *n* (%)	3 (7.3)	3 (5.7)	0.745
Surgical resection, *n* (%)	3 (7.3)	4 (7.5)	0.966
Ablative procedures, *n* (%)	7 (17.1)	3 (5.7)	0.075
TARE or TACE, *n* (%)	4 (9.8)	7 (13.2)	0.606
Systemic therapies, *n* (%)	6 (14.6)	1 (1.9)	**0.020**
Liver transplantation, *n* (%)	1 (2.4)	0 (0.0)	0.253

* *t* value and degree of freedom (df) is reported for only *t*-test results. INR, international normalized ratio; TARE, transarterial radioembolization; TACE, transarterial chemoembolization.

**Table 3 medicina-58-01305-t003:** The 1-, 3-, and 5-year disease-free survival rates of patients who underwent anatomic and non-anatomic liver resections for treatment of hepatocellular carcinoma (HCC).

					95% CI	
Levels	Time	Number at Risk	Number of Events	Survival	Lower	Upper
Non-anatomic	12	39	14	73.6 %	62.6 %	86.5 %
	36	13	17	39.1 %	27.5 %	55.5 %
	60	8	2	32.8 %	21.4 %	50.3 %
Anatomic	12	20	21	48.8 %	35.6 %	66.8 %
	36	6	10	22.7 %	12.6 %	40.9 %
	60	5	0	22.7 %	12.6 %	40.9 %

CI, confidence interval.

**Table 4 medicina-58-01305-t004:** The 1-, 3-, and 5-year overall survival rates of patients who underwent anatomic and non-anatomic liver resections for treatment of HCC.

					95% CI	
Levels	Time	Number at Risk	Number of Events	Survival	Lower	Upper
Non-anatomic	12	43	10	81.1 %	71.3 %	92.4 %
	36	22	12	56.6 %	44.4 %	72.3 %
	60	13	4	44.5 %	31.7 %	62.6 %
Anatomic	12	33	8	80.5 %	69.2 %	93.6 %
	36	19	8	59.1 %	45.4 %	76.9 %
	60	13	3	49.3 %	35.3 %	68.9 %

CI, confidence interval.

## Data Availability

The data that support the findings of this study are available from the corresponding author upon reasonable request.

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
