# Peer review of "Comparison of Anatomic and Non-Anatomic Liver Resection for Hepatocellular Carcinoma: A Retrospective Cohort Study"

_medicina, 2022, doi:10.3390/medicina58091305_

Round 1

Reviewer 1 Report

 The article looks interesting but needs modification.  The title needs to be modified..no need for a mention of a review of literature as it is purely a research article...The data need to be explained in graphs sometimes. The figure legends/table legends should be explained.  The authors should give much attention to improving the quality of data presentation...If possible a few more pic related to the study can be captured

 Please see the model article for reference

Front Public Health. 2021; 9: 816704. Published online 2022 Feb 8. doi: 10.3389/fpubh.2021.816704 PMCID: PMC8863048 PMID: 35211454

Comparison of the Efficacy of Anatomic and Non-anatomic Hepatectomy for Hepatic Alveolar Echinococcosis: Clinical Experience of 240 Cases in a Single Center

Author Response

Dear Reviewer,

We pleased to receive your comments and tried our best to present our study to deserve publication in Medicina. We applied comments in the manuscript, added a graph, an operation picture and an illustration, and reviewed our statistic output files. Changes and responses are listed below. We hope you to find them sufficient.

Kind regards 

Reviewer 2 Report

This article is interesting and discusses a hot topic in the field of hepatology, but I have comments that are mentioned below:

1- it is a retrospective study.

2- multinational metanalysis is needed to prove the results of this study and the same studies in this field.

3- even if in BCLC, class d, surgery is still working and this is fruitful to those patients, so why do multicenter large population studies don't discuss this issue?

4- for each p-value please add a degree of freedom.

5-what is the situation when there are portal vein Mets? Do you proceed with surgery or it is a relative contraindication to the procedure?

Author Response

(The authors gave the same response as above.)

Round 2

Reviewer 1 Report

The authors can improve the figure and table legends for more clarity.

confirm that all the figures are not taken from a third party.
